# Immunogenicity and Protective Capacity of a Virus-like Particle Vaccine against *Chlamydia trachomatis* Type 3 Secretion System Tip Protein, CT584

**DOI:** 10.3390/vaccines10010111

**Published:** 2022-01-12

**Authors:** Everett Webster, Kyra W. Seiger, Susan B. Core, Amanda L. Collar, Hannah Knapp-Broas, June Graham, Muskan Shrestha, Sarah Afzaal, William M. Geisler, Cosette M. Wheeler, Bryce Chackerian, Kathryn M. Frietze, Rebeccah S. Lijek

**Affiliations:** 1Department of Biological Sciences, Mount Holyoke College, 50 College St., South Hadley, MA 01075, USA; webst22e@mtholyoke.edu (E.W.); seige22k@mtholyoke.edu (K.W.S.); knapp22h@mtholyoke.edu (H.K.-B.); graha22j@mtholyoke.edu (J.G.); shres25m@mtholyoke.edu (M.S.); afzaa22s@mtholyoke.edu (S.A.); 2Department of Molecular Genetics and Microbiology, School of Medicine, University of New Mexico, MSC 08-4660, 1 University of New Mexico, Albuquerque, NM 87131, USA; score@salud.unm.edu (S.B.C.); collaral@salud.unm.edu (A.L.C.); bchackerian@salud.unm.edu (B.C.); kfrietze@salud.unm.edu (K.M.F.); 3Department of Medicine, University of Alabama at Birmingham, 703 19th St. S, ZRB 242, Birmingham, AL 35294, USA; wgeisler@uabmc.edu; 4Center for HPV Prevention, University of New Mexico Comprehensive Cancer Center, University of New Mexico Health Sciences Center, MSC 08-4640, 1 University of New Mexico, Albuquerque, NM 87131, USA; cwheeler@salud.unm.edu; 5Clinical and Translational Science Center, University of New Mexico Health Sciences, MSC 08-4635, 1 University of New Mexico, Albuquerque, NM 87131, USA

**Keywords:** *Chlamydia trachomatis*, virus-like particle vaccines, type 3 secretion system

## Abstract

An effective vaccine against *Chlamydia trachomatis* is urgently needed as infection rates continue to rise and *C. trachomatis* causes reproductive morbidity. An obligate intracellular pathogen, *C. trachomatis* employs a type 3 secretion system (T3SS) for host cell entry. The tip of the injectosome is composed of the protein CT584, which represents a potential target for neutralization with vaccine-induced antibody. Here, we investigate the immunogenicity and efficacy of a vaccine made of CT584 epitopes coupled to a bacteriophage virus-like particle (VLP), a novel platform for *Chlamydia* vaccines modeled on the success of HPV vaccines. Female mice were immunized intramuscularly, challenged transcervically with *C. trachomatis,* and assessed for systemic and local antibody responses and bacterial burden in the upper genital tract. Immunization resulted in a 3-log increase in epitope-specific IgG in serum and uterine homogenates and in the detection of epitope-specific IgG in uterine lavage at low levels. By contrast, sera from women infected with *C. trachomatis* and virgin controls had similarly low titers to CT584 epitopes, suggesting these epitopes are not systemically immunogenic during natural infection but can be rendered immunogenic by the VLP platform. *C. trachomatis* burden in the upper genital tract of mice varied after active immunization, yet passive protection was achieved when immune sera were pre-incubated with *C. trachomatis* prior to inoculation into the genital tract. These data demonstrate the potential for antibody against the T3SS to contribute to protection against *C. trachomatis* and the value of VLPs as a novel platform for *C. trachomatis* vaccines.

## 1. Introduction

*Chlamydia trachomatis* is the most common sexually transmitted bacterial pathogen with an annual global incidence rate estimated at 127 million cases in 2016 [1]. Although infection can be treated with antibiotics, an estimated 70–80% of *C. trachomatis* infections in women are asymptomatic and so can go untreated without routine surveillance [2]. As a result, 15–40% of *C. trachomatis* infections in women ascend to the upper genital tract and trigger immunopathology that leads to pelvic inflammatory disease (PID), tubal infertility, and increased risk for ectopic pregnancies [2,3,4]. *C. trachomatis*-infected individuals are also observed to be at increased risk of contracting HIV and other STIs, in part due to damage and inflammation of the genital epithelium [4]. Despite aggressive treatment and control efforts, *C. trachomatis* infection rates are increasing, which underscores the need to develop a *C. trachomatis* vaccine as described recently by the World Health Organization (WHO) and the US National Institute of Allergy and Infectious Diseases (NIAID) [5,6].

Vaccine development against *C. trachomatis* spans a century and a variety of approaches (recently reviewed in [7]), yet no *C. trachomatis* vaccines are currently available. A phase I clinical trial of a recombinant subunit vaccine against immunodominant antigen MOMP (“CTH522”) was recently completed, representing the first human trial of a *C. trachomatis* vaccine in 50 years [7,8]. MOMP is the major outer membrane protein of *C. trachomatis* and the most extensively investigated *C. trachomatis* vaccine antigen to date [9]. Experimental vaccination with recombinant MOMP has had variable success, sometimes attributed to the lack of native protein morphology [7,10]. Current efforts have shifted to using smaller regions of MOMP, especially a conserved epitope centered within variable domain 4 (VD4) [10]. In a clinical trial, the MOMP-VD4 CTH522 vaccine was shown to be safe and immunogenic in 15 women, inducing neutralizing antibody in serum after three intramuscular injections, though MOMP-specific IgG and IgA were less prevalent in mucosal secretions and not shown to be neutralizing [8]. As with vaccination with CTH522, genital infection with *C. trachomatis* induces a robust antibody response against MOMP-VD4 [11,12]. However, this response is not protective: re-infection with the same *C. trachomatis* serovar is common [13] and not associated with the quantity or quality of the MOMP-VD4 antibody response [12]. Whether vaccination with CTH522 protects against infection with *C. trachomatis* will need to be determined by further clinical trials.

Here, we investigate a novel approach to *C. trachomatis* vaccination informed by the success of vaccines against another sexually transmitted pathogen: human papillomavirus (HPV). Both HPV vaccines, Gardasil (Merck) and Cervarix (GSK), use non-infectious HPV virus-like particles (VLPs) to stimulate robust protection from genital tract infection and disease, driven by the production of high-titer neutralizing antibodies [14,15,16,17,18,19]. HPV VLP vaccines elicit HPV-specific IgG in serum and the genital tract [15,20] that is sufficient to block HPV binding to epithelial cells [21] and induce sterilizing immunity in animal models [17,18,19]. As such, the HPV vaccines provide proof of principle that VLPs are a vaccine platform capable of producing high titer antibody and protection from infection in the female genital tract [22]. We reasoned that a VLP platform displaying *C. trachomatis* antigen(s) could also generate high titer antibodies with the potential to protect against *C. trachomatis* infection in the female genital tract [22]. The self-adjuvating immunogenicity of VLPs can be directed toward exogenous peptides of interest that are chemically conjugated to the outside of the particle [23,24]. To date, VLP vaccines targeting *C. trachomatis* antigens have not been explored [7].

Rationally, an ideal antigen to target with vaccine-induced antibody would be surface-exposed on the pathogen and crucial to entry or infection in the host. The type 3 secretion system (T3SS) is a critical virulence factor for many Gram-negative bacterial pathogens, including *C. trachomatis*, and is being explored as an antigen for experimental vaccines against *Shigella*, *Salmonella*, *Yersinia*, and *Pseudomonas* [25,26,27,28]. The T3SS is present during all stages of the biphasic life cycle of *C. trachomatis* and is conserved across *C. trachomatis* serovars and other *Chlamydia* species [2]. In the first step of infection, contact between the *C. trachomatis* elementary body (EB) and a host epithelial cell leads to the secretion of effector proteins from the T3SS injectosome, triggering the recruitment and reorganization of actin and facilitating engulfment of the EB into the host cell cytosol [29]. The tip of the injectosome in the *C. trachomatis* T3SS is predicted to be composed of CT584 [30,31], making CT584 a rational target for VLP-induced antibody-mediated inhibition of the T3SS and protection from infection. CT584 was included in a trivalent fusion protein vaccine which reduced vaginal shedding and hydrosalpinx from *C. muridarum* [32], suggesting it may also prove useful in protecting against *C. trachomatis*. The value of CT584 alone in vaccines against *C. trachomatis* has not yet been explored.

Here, we construct and evaluate a novel bacteriophage VLP vaccine against *C. trachomatis* CT584 epitopes. Mice were immunized intramuscularly, challenged transcervically with *C. trachomatis,* and assessed for systemic and local antibody responses and bacterial burden in the upper genital tract, the site of disease in humans. Sera from immunized mice were also pre-incubated with *C. trachomatis* prior to infection to determine whether vaccine-induced antibody against CT584 is sufficient to reduce infection in vivo. Together, these data inform future vaccine development by providing the first in vivo evaluation of a VLP vaccine against *C. trachomatis*.

## 2. Materials and Methods

**Synthesis of virus-like particle vaccines**. Each VLP vaccine consisted of bacteriophage Qβ capsid chemically crosslinked to peptides predicted to be B cell epitopes for *C. trachomatis* antigen CT584. Qβ VLPs were produced in *Escherichia coli*, as previously described [33]. Recombinant Qβ capsid protein self-assembles into an icosahedral capsid with t = 3 symmetry consisting of 90 protein dimers with surface exposed lysine residues. The peptides selected to be vaccine antigens were synthesized to include a C-terminus cysteine residue preceded by three glycines as a spacer (Gen-Script, Piscataway, NJ, USA) and conjugated to surface-exposed lysines on Qβ VLPs using a bifunctional cross-linker, succinimidyl 6-[(β-maleimidopropionamido) hexanoate] (Thermo Scientific, Waltham, MA, USA). Conjugation was confirmed by denaturing polyacrylamide gel electrophoresis. Each peptide-conjugated VLP was mixed in equal ratio in 1× phosphate buffered saline (PBS) to create the final immunization mixture.

**Mice and immunization.** All mouse procedures were performed in accordance with protocols approved by the Institutional Animal Care and Use Committees of Mount Holyoke College (#BR-57-1117a) and the University of New Mexico (#19-200867). Female C57BL/6 mice were purchased from The Jackson Laboratory (USA), housed at Mount Holyoke College, and provided food and water ad libitum. C57BL/6 mice were used for transcervical *C. trachomatis* infections in accordance with previous studies that use the female upper genital tract *C. trachomatis* infection model [34,35] and investigate the immunogenicity of *C. trachomatis* peptide antigens [36]. Mice used in vaccine trials were three weeks old at the start of the three month-long experiment. Mice for all other experiments were used at six to eight weeks of age. Mice were immunized intramuscularly in the hind leg with 5 μg of either Qβ alone (negative control) or Qβ conjugated to *C. trachomatis* epitope(s) of interest. Serum was collected via tail vein prior to immunization, after immunization and before challenge, and after challenge at the time of sacrifice by cardiac bleed. The upper genital tract was excised and mechanically homogenized for quantification of bacterial burden, see quantification methods below.

To collect uterine lavage, female Balb/c mice housed at the University of New Mexico were immunized as described above. After sacrifice and exsanguination, the uterus was excised and washed externally in 1 mL of 1× PBS. Then 200 μL of fresh 1× PBS was instilled into the uterus transcervicaly and the resulting lavage was collected via the uterine horns. 

**Bacterial challenge and quantification.** *Chlamydia trachomatis* serovar D (UW-3/Cx, ATCC, Manassas, VA, USA) was cultured, inoculated into mice, and quantified as previously described [34]. Briefly, bacteria were propagated within McCoy cells (ATCC, Manassas, VA, USA) and released from the inclusion using sterile glass beads and sonication. Elementary bodies (EBs) were purified by density gradient centrifugation, stored at −80 °C, thawed immediately before use, and titered on McCoy cell monolayers to quantify inclusion forming units (IFU). 5 × 10^6^ IFU of purified EBs were inoculated trans-cervically into the upper genital tract of female mice using a nonsurgical embryo transfer device (ParaTechs, Lexington, KY, USA). Mice were treated subcutaneously with 2.5 mg medroxyprogesterone acetate (Pfizer, New York, NY, USA) one week before infection and sacrificed three days post infection. Bacterial burden in the upper genital tract was determined using quantitative PCR, shown previously to accurately reflect IFU [35]. Total DNA was isolated from homogenized upper genital tracts using a DNeasy Blood and Tissue Kit (Qiagen, Beverly, MA, USA). *C. trachomatis* 16S DNA and mouse GAPDH DNA was amplified and quantitated on an AriaMX Real-Time PCR System (Agilent, Santa Clara, CA, USA) using specific primer pairs and probes (IDT, Coralville, IA, USA or Applied Biosystems, Waltham, MA, USA). The ratio of *C. trachomatis* to mouse DNA was calculated using standard curves generated from known amounts of purified *C. trachomatis* or mouse DNA. 

**Pre-opsonization.** Pre-opsonization of *C. trachomatis* was performed as previously described [8]. Serum samples were heat-inactivated for 30 min at 56 °C, cooled, then incubated at 37 °C for 30 min with purified *C. trachomatis* EBs at a 1:1 volume ratio to allow for pre-opsonization immediately prior to transcervical infection.

**Antibody quantification by ELISA**. Peptide-specific and/or total IgG was quantified in serum, upper genital tract homogenates, and uterine lavage by ELISA. For peptide-specific ELISA, Immulon 2 HB plates (Thermo Scientific, Waltham, MA, USA) were coated in 0.5 μg of streptavidin (Invitrogen, USA), washed in 1× PBS (Thermo Scientific, USA), coated in 1 μg of the crosslinker succinimidyl 6-[(β-maleimidopropionamido) hexanoate] (Thermo Scientific, USA), washed again in 1× PBS, then coated with 1 μg of peptide. Plates were blocked with 0.5% dry milk (Quality Biological, Gaithersburg, MD, USA) in 1× PBS then incubated with serial dilutions of each sample (serum, homogenate, or lavage). Secondary antibody conjugated to horseradish peroxidase (Peroxidase AffiniPure goat anti-mouse IgG, Jackson ImmunoResearch, West Grove, PA, USA) was added at a 1:5000 dilution and detected by TMB substrate (Thermo Scientific, USA) quenched with 1% HCl prior to reading the optical density at 450 nm (OD450). Endpoint dilution was defined as the reciprocal of the highest sample dilution with an OD450 reading twice that of the blank wells. For the total IgG ELISA, a serial dilution of uterine lavage was added directly to Immulon 2 HB plates and the remainder of the ELISA steps were performed as above. OD450 values were reported for 1:2 dilution of uterine washes by subtracting the background value from the given value.

**Human serum samples.** Collection and use of human serum samples was approved as previously described [12] by the University of Alabama at Birmingham Institutional Review Board, the Jefferson County Department of Health (JCDH) in Birmingham, AL, USA, and the University of New Mexico Health Sciences Center Human Research Review Committee. Participants were enrolled from two populations: (1) predominantly African-American *C. trachomatis*-infected women aged >16 years presenting to the JCDH STD Clinic in Birmingham, AL, USA, and (2) Hispanic and non-Hispanic white virgin women aged 18–40 years presenting for routine gynecologic examinations in Albuquerque, NM, USA. All study participants provided informed consent. For the *C. trachomatis*-infected cohort, *C. trachomatis* infection was confirmed by nucleic acid amplification testing (Hologic Aptima Combo 2; Hologic, Inc., Marlborough, MA, USA) and exclusion criteria included known co-infection with gonorrhea, syphilis, or HIV, as well as pelvic inflammatory disease. From both cohorts, blood samples were obtained for serum, which was used in ELISAs to measure IgG titers to MOMP VD4 (as was shown in [12] and presented here for comparison, *n* = 7 virgin, *n* = 20 *C. trachomatis*-infected women) and to CT584 epitopes 70–77 and 154–164 (*n* = 17 virgin, *n* = 20 *C. trachomatis*-infected women).

**Statistical analysis**. Two-to-three independent experiments were performed for each analysis, with at least five mice per group. Statistical analysis was performed using Prism (GraphPad, San Diego, CA, USA) and a *p* value of <0.05 was considered significant, where **** *p* < 0.0001, *** *p* < 0.001, ** *p* < 0.01, * *p* < 0.05, and n.s. indicates not significant by Mann Whitney U test unless otherwise indicated. The nonparametric Mann Whitney U test was selected for pairwise comparisons with a small sample size where the data did not follow a normal distribution.

## 3. Results

### 3.1. Creation of a VLP Vaccine against C. trachomatis CT584

The goal of this study was to develop a *C. trachomatis* vaccine that employs a bacteriophage virus-like particle (VLP) platform and assess its ability to elicit antibody and protect against female upper genital tract infection. For the VLP, we selected the capsid from the bacteriophage Qβ, which was shown in clinical trials to be well-tolerated and to elicit robust antibody responses to the peptide conjugated to its surface without the need for adjuvant [37,38]. To identify potential vaccine epitopes, the amino acid sequence of CT584 from *C. trachomatis* serovar D (#AAC68186.1) was analyzed using the Immune Epitope Database (IEDB) [39]. Characteristics including hydrophilicity, flexibility, accessibility, turns, exposed surface, polarity, and antigenicity were used to determine linear epitopes conducive to B-cell recognition and binding. Using linear epitopes avoids the challenges with native protein conformation seen with other experimental *C. trachomatis* vaccines [7]. The two most optimal B-cell epitopes of CT584 identified by IEDB were residues 70–77 and 154–164, both of which are surface-exposed on the outward face of the CT584 hexamer that assembles at the tip of the T3SS (Figure 1). CT584 peptides 70–77 and 154–164 were synthesized with a C-terminus spacer to allow for chemical conjugation to the Qβ VLP (Figure 1A). Conjugation of each peptide to the Qβ VLP was confirmed via denaturing polyacrylamide gel electrophoresis. VLPs conjugated separately with peptides 70–77 and 154–164 were then combined at a 1:1 ratio to create the mixed vaccine formulation, Qβ-CT584. Unconjugated Qβ VLPs lacking any *C. trachomatis* epitopes were used as the negative control. 

### 3.2. Urogenital C. trachomatis Infection in Women Does Not Elicit Antibody to Qβ-CT584 Vaccine Epitopes

Having identified potential vaccine epitopes for CT584, we next investigated whether natural genital tract infection with *C. trachomatis* in women resulted in antibodies to these epitopes. We predicted that these CT584 epitopes would not be immunogenic during natural infection since it is advantageous for *C. trachomatis* to evade immunity to critical virulence factors such as the T3SS. Indeed, IgG specific for CT584 epitopes 70–77 or 154–164 was similarly low in virgin and *C. trachomatis*-infected women (Figure 2). Increases in antibody levels to CT584 70–77 or 154–164 after *C. trachomatis* infection were minimal compared to the 5.45 fold increase to the immunodominant antigen MOMP (Figure 2) [12]. One interpretation of these data is that CT584 70–77 and 154–164 are not robustly targeted during natural *C. trachomatis* infection and so have the potential to be cryptic epitopes if protective antibody can be elicited by a CT584-directed vaccine.

### 3.3. Immunization with Qβ-CT584 Induces High Titer, Peptide-Specific Serum IgG 

We hypothesized that CT584 70–77 and 154–164 peptide epitopes could be made immunogenic by display on a VLP platform. To investigate the antibody response to CT584 epitopes when incorporated into VLP vaccines, female C57Bl/6 mice were immunized intramuscularly with either the single epitope vaccines Qβ-70 or Qβ-154, or the mixed formulation Qβ-CT584, or the unconjugated negative control Qβ three times at three-week intervals. This follows an immunization scheme previously developed for other Qβ VLP vaccines [33]. The titer of serum IgG specific to CT584 peptides 70–77 and 154–164 was measured by ELISA after immunization and prior to *C. trachomatis* challenge (Figure 3A). Immunization with both the single epitope vaccines and the mixture Qβ-CT584 elicited a robust serum IgG response to the CT584 peptides, with titers 10,000-fold greater than the negative control. No CT584-specific antibody was detected in mice immunized with Qβ alone. Antibody titers to Qβ-CT584, the mixed epitope formulation with 50% of the dosage of each VLP compared to the single epitope vaccines, were similarly high as the single epitope formulations. These results underscore the value of Qβ as a highly immunogenic VLP platform, consistent with previous research [33]. They also demonstrate that CT584 epitopes, which are not immunogenic during natural infection, have the potential to be highly immunogenic when incorporated into a VLP vaccine.

### 3.4. Protection from C. trachomatis Infection after Immunization with Qβ-CT584 Varies

Mice immunized with the single epitope vaccines Qβ-70 or Qβ-154, the mixed formulation vaccine Qβ-CT584, or the negative control Qβ alone were challenged trans-cervically with *C. trachomatis* to determine if vaccination reduced bacterial burden in the upper genital tract (Figure 3B). Bacterial burden in uterine homogenates was quantified by qPCR three days after challenge. Immunization with Qβ-CT584 resulted in a statistically significant 92% reduction in *C. trachomatis* burden compared to Qβ alone (mean of 5.5 vs. 66.8 pg 16 S/μg GAPDH DNA, *p* = 0.0079, Figure 3B). Immunization with the single epitope vaccine Qβ-154 or Qβ-70 was not sufficient to significantly reduce bacterial burden (65% and 4% reduction, respectively), highlighting the value in targeting multiple epitopes simultaneously with peptide vaccines.

Based on these data, we performed follow-up analyses on the mixed epitope vaccine, Qβ-CT584. Immunization with Qβ-CT584 or Qβ alone followed by transcervical *C. trachomatis* challenge was performed as before, this time with an increased sample size of 20 mice per group as determined by a power analysis (Figure 4). Larger group sizes revealed more within-group variability and a loss of statistically-significant protective capacity (Figure 4A). The failure to achieve infection in four of the Qβ-immunized control animals may have contributed to these findings. Epitope-specific IgG titers in uterine homogenates also varied yet on average were three logs higher than negative controls (Figure 4B). Taken together, *C. trachomatis* burden was reduced in 72% of the mice immunized with Qβ-CT584 compared to the mean of the Qβ-immunized negative control group. 

### 3.5. Vaccine-Induced Antibody Is Detectable at the Site of Infection

To investigate whether high bacterial burden could be caused by a lack of vaccine-induced antibody at the site of *C. trachomatis* infection, the uterine epithelium, we lavaged the lumen of uteri from exsanguinated mice immunized with Qβ-CT584 or Qβ alone. Vaccine epitope-specific IgG was compared by ELISA in lavage versus serum (Figure 5). As before, vaccination with Qβ-CT584 induced high titer serum antibody to vaccine epitopes (Figure 5A). Epitope-specific antibody in uterine lavage was detectable, yet at levels 4 logs lower than those in serum (Figure 5B). Total IgG was equivalent in lavage from Qβ-CT584 and Qβ immunized mice (Figure 5C). Given that lavage is inherently a dilution of mucosal antibody, it is difficult to make a direct comparison between lavage and serum titers. Regardless, parenteral immunization with this VLP platform is capable of inducing a detectable mucosal antibody response, though functionally this response may be suboptimal to mediate robust protection.

### 3.6. Pre-Incubation with Qβ-CT584-Immune Sera Reduces C. trachomatis Burden in the Upper Genital Tract

We sought to determine whether vaccine-induced antibody against the T3SS could be sufficient to inhibit *C. trachomatis* infection in vivo when made available at high titer. *C. trachomatis* EBs were pre-opsonized with heat-inactivated immune sera pooled from mice vaccinated with Qβ-CT584 or the negative control Qβ alone. Pre-opsonized *C. trachomatis* Ebs were then trans-cervically inoculated into naive mice. Bacterial burden was measured by qPCR three days after infection. Pre-incubation with Qβ-CT584-immune serum was sufficient to inhibit *C. trachomatis* infection in the upper genital tract (Figure 6). Bacterial burden was reduced by 84% after pre-incubation with Qβ-CT584-immune sera from the *n* = 5 vaccine trial where protection was observed (Figure 6A, serum from Figure 3B). When sera from the *n* = 20 vaccine trial (Figure 4A) were used for pre-opsonization, sera from the five mice with the lowest bacterial burden were pooled to test proof of principle and reduce within-group variability. Pre-opsonization with these Qβ-CT584 immune sera resulted in a similar 83% reduction in *C. trachomatis* burden (Figure 6B). Together, these data demonstrate that vaccination with a VLP vaccine against *C. trachomatis* T3SS tip protein CT584 elicits serum antibody that has the capability to inhibit *C. trachomatis* infection in the upper genital tract.

## 4. Discussion

The WHO and NIAID have called for further research to develop *Chlamydia trachomatis* vaccine platforms, identify protective antigens, and to elucidate the role that antibody might play in prophylactic immunization [5,6]. This study moves *C. trachomatis* vaccine development forward by combining a novel platform, Qβ virus-like particles, with an understudied yet promising antigen, the T3SS tip protein CT584, and evaluating their immunogenicity and efficacy in the female upper genital tract. The Qβ-CT584 vaccine elicited IgG to both of the CT584 B-cell epitopes crosslinked to the VLP, adding to the evidence that Qβ VLPs are an immunogenic platform even in the absence of adjuvant. Epitope-specific titers were high in serum, variable in uterine homogenates, and low but detectable in uterine lavage. Protection from active immunization was variable and likely driven by serum antibody, since passive protection was repeatedly observed when immune sera were pre-incubated with *C. trachomatis* prior to genital tract infection. These data demonstrate the importance of the T3SS in establishing *C. trachomatis* infection in the uterus and the potential for VLP vaccine-induced antibody against the T3SS to contribute to protection. Although active immunization with these CT584 epitopes yielded equivocal results, ongoing *C. trachomatis* vaccine development would benefit from further exploration of VLP platforms.

The selection of a VLP platform for our *C. trachomatis* vaccine, Qβ-CT584, was inspired by the success of the VLP-based HPV vaccines, the only licensed vaccines capable of preventing sexually-transmitted infection and disease in the female genital tract. Although the HPV VLP vaccines use native HPV capsid compared to the bacteriophage capsid used here, both types of VLP vaccines protect through the production of high-titer, epitope-specific antibody, achieved through the highly repetitive antigenic structure of the particle [16]. As a result, we focused the design and evaluation of our VLP vaccine on humoral immunity rather than cell-mediated immunity. Long-term protective immunity against *C. trachomatis* may well require stimulation of both humoral and cell-mediated immunity. Immunity in the genital tract could potentially be enhanced by using a mucosal immunization strategy or a heterologous “prime-pull” approach as is being pursued for herpes simplex virus [40]. *C. trachomatis* vaccines may similarly benefit from a combination of administration routes and/or heterologous antibody and T-cell stimulating features. Peptide-conjugated bacteriophage VLPs, such as those we explore in this study, are potent antibody-stimulators that may be useful as components in future *Chlamydia* immunization schemes.

Prior to the proven success of the VLP HPV vaccines, it was unclear whether antibody would be effective against HPV since it did not contribute to protection during natural infection [41]. Similarly, antibody elicited by natural infection has been shown to be dispensable during the clearance of primary upper genital tract infection with *C. trachomatis* in mice [35] and in women does not correlate with re-infection outcomes [12]. Yet, others have shown that protective antibody responses superior to those from natural infection can be achieved exogenously through monoclonal antibody treatments or experimental vaccines, as with the HPV vaccines [10,42,43,44]. Our data add to this line of evidence by demonstrating that pre-incubation with high-titer Qβ-CT584 immune serum is sufficient to reduce *C. trachomatis* burden in the upper genital tract. Antibody could reduce bacterial burden by neutralizing the bacteria (i.e., hindering attachment or virulence factor function) or through enhancing opsonophagocytic clearance of the bacteria. The use of an in vivo protection assay (in Figure 6) instead of an in vitro neutralization assay allows us to be inclusive of both effector functions, which could be teased apart in further mechanistic studies. Antibody may also impact time-to-clearance which could be assessed in future time course analyses using *C. trachomatis* or the *C. muridarum* vaginal infection model.

Vaccination against the T3SS is a common approach being developed for several pathogenic Gram-negative bacteria [25,26,27,28]. Our study, which targets *C. trachomatis* T3SS tip protein CT584, adds to the growing evidence that the injectosome tip is a useful antigen for antibody-mediated immunotherapy. As we show for *C. trachomatis*, antibody against the T3SS tip protein of other bacterial pathogens is particularly useful in generating passive protection: passive antibody blockade of the *Pseudomonas aeruginosa* T3SS tip protein, PcrV, was shown to be protective in multiple small animal models [45,46,47,48,49,50]. Progress has also been made with targeting the T3SS injectosome tip of *Chlamydia* spp., often in multivalent formulations. The T3SS tip protein from *C. muridarum*, TC0873, was included in a hexavalent subunit vaccine that reduced *C. muridarum* vaginal shedding and hydrosalpinx after intranasal immunization [51]. Immunization with a fusion protein of T3SS tip protein CT584 with translocator proteins CopB and CopD also reduced *C. muridarum* vaginal shedding at early times post infection and decreased the incidence of hydrosalpinx in 5 mice when delivered intranasally with CpG adjuvant [32]. Our study adds to this literature by showing that antibody against two epitopes from T3SS tip protein CT584 alone is sufficient to passively reduce *C. trachomatis* burden three days post infection. These data together advance our understanding of the potential value of targeting the T3SS tip with antibody to inhibit *Chlamydia* genital tract infection.

The complexity of previous *Chlamydia* T3SS vaccines make it difficult to ascertain the relative contribution of each antigen and to define the epitopes required for antibody-mediated protection. Here, we take a reductionist approach and target only two linear B-cell epitopes, amino acids 70–77 and 154–164 of CT584. A limitation of this approach is that these highly specific and minimal peptides may be insufficient on their own to confer robust protection after active immunization, especially in mouse models where logarithms of variability in bacterial burden is not uncommon [10,32,51]. Variability may be more pronounced in immunization experiments when the advanced age of the mice at challenge is out of the ideal infectivity window of six to eight weeks of age. The cause of this variability is unknown and whether it is physiologically representative of the variable outcomes of human *C. trachomatis* infection (i.e., asymptomatic carriage versus PID) is an interesting and open question. 

A major strength of peptide vaccines is that they do not cause off-target pathological effects, which have historically plagued *Chlamydia* vaccine development [7]. Importantly, they also focus the immune response to known protective epitopes, such as that within the MOMP variable domain 4 currently targeted by CTH522 in clinical trials [10]. The exact VD4 neutralizing epitope is not found in *C. muridarum* [10], underscoring the value in employing *C. trachomatis* in preclinical vaccine studies as was carried out here. Peptide vaccines are useful for directing the immune response away from “immunological decoys”, often unconserved and immunodominant epitopes that drive antibody to nonessential and thus non-protective regions of the protein [52]. Instead, peptide vaccines can reveal cryptic epitopes, which are not immunogenic during natural infection but could be protective when targeted by vaccines. Whether CT584 epitopes 70–77 and 154–164 are cryptic epitopes remains to be determined; natural infection in women did not produce robust serum IgG, but these systemic titers may not be representative of mucosal antibody titers. Immunity to mucosal infection and/or immunization is likely to differ from intramuscular immunization with these peptides, and there may also be species-specific differences in immunogenicity between humans and mice. Future studies of the genital mucosal immune response to CT584 epitopes in humans and mouse models are warranted. Cryptic epitopes are being explored for immunotherapies against other mucosal pathogens such as group A streptococcus [53], the L2 protein of HPV [54], HIV [55] and SARS-CoV-2 [56]. Using VLPs to target cryptic epitopes from the T3SS or other essential virulence factors may benefit the ongoing development of effective vaccines against *C. trachomatis*.

## Figures and Tables

**Figure 1 vaccines-10-00111-f001:**
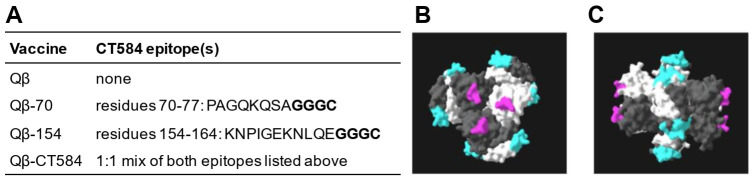
Development of virus-like particle vaccines against *C. trachomatis* T3SS tip protein CT584. (**A**) Linear B-cell epitopes of *C. trachomatis* CT584 were identified by the Immune Epitope Database and used in VLP vaccines. Bolded residues were added for chemical conjugation to the VLP composed of the capsid from the bacteriophage Qβ. (**B**,**C**) Localization of vaccine epitopes 70–77 (pink) and 154–164 (cyan) in the CT584 hexamer from the top (**B**) and side (**C**) view. Structure retrieved from NCBI PDB #4MLK.

**Figure 2 vaccines-10-00111-f002:**
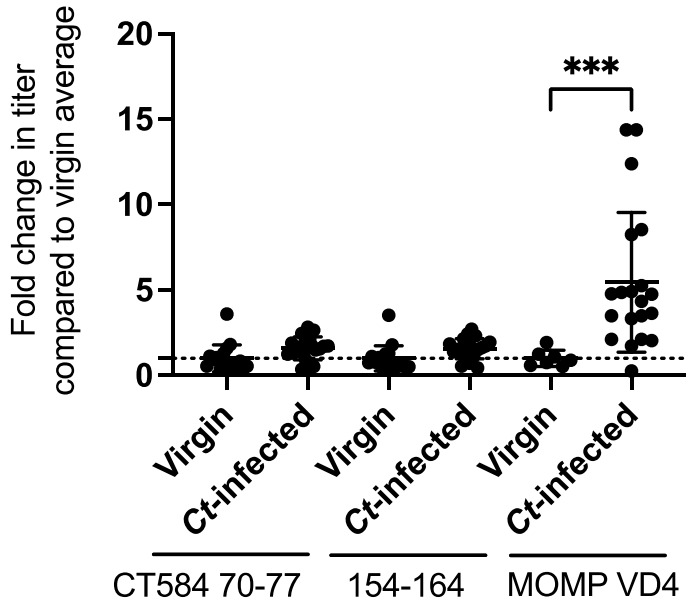
Urogenital tract infection with *C. trachomatis* in women does not elicit robust antibody responses to Qβ-CT584 vaccine epitopes. Virgin and *C. trachomatis* (*Ct*)-infected women were evaluated for serum IgG to CT584 epitopes 70-77 and 154-164, as well as to the immunodominant *C. trachomatis* epitope MOMP VD4. Genital tract infection with *C. trachomatis* resulted in an average 5.45 fold increase in MOMP VD4 titers (*** *p* < 0.001, data adapted from [12]) compared to an average 1.58 fold and 1.56 fold increase to CT584 epitopes 70–77 and 154–164, respectively.

**Figure 3 vaccines-10-00111-f003:**
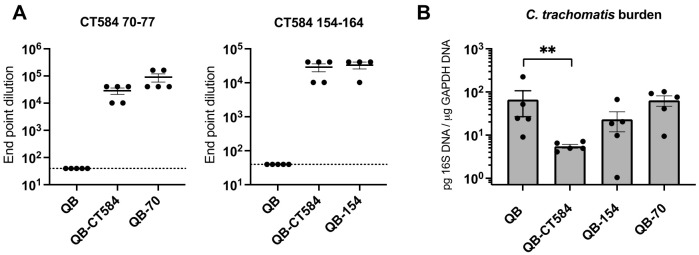
Immunogenicity and protective capacity of single epitope and mixed epitope vaccines against *C. trachomatis* T3SS tip protein CT584. (**A**) Immunization with Qβ VLPs conjugated to CT584 epitopes 70–77 (Qβ-70), 154–164 (Qβ-154) or a mixture of both (Qβ-CT584) induced high titer IgG specific to epitopes 70–77 (**left**) and 154–164 (**right**) as measured by ELISA. End point dilution was defined as the reciprocal of the highest sample dilution with an OD450 reading twice that of blank wells. Horizontal lines represent means +/− standard error of the mean (SEM), dotted lines represent lower limit of detection. (**B**) Bacterial burden in the upper genital tract of female mice immunized with either Qβ-CT584, Qβ-70, Qβ-154 or the negative control Qβ alone was measured by qPCR. Immunization with Qβ-CT584 resulted in a 92% reduction in *C. trachomatis* load, ** *p* = 0.0079 by Mann Whitney U test.

**Figure 4 vaccines-10-00111-f004:**
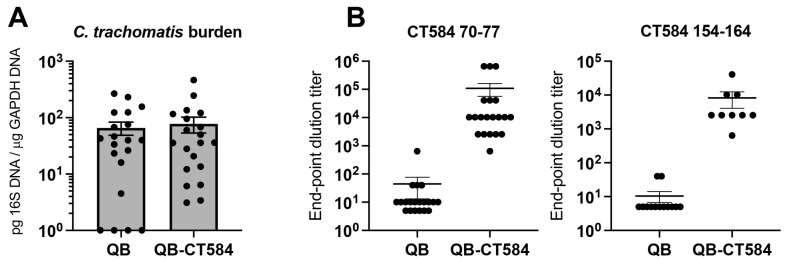
Immunogenicity and protective capacity of vaccine Qβ-CT584. A second, statistically-powered vaccine trial (*n* = 20/group) was performed to assess the protective capacity and immunogenicity of Qβ-CT584. (**A**) Bacterial burden in the upper genital tract of female mice immunized with either Qβ-CT584 or the negative control Qβ alone was measured by qPCR. (**B**) Immunization with Qβ-CT584 induced high titer IgG specific to vaccine epitopes CT584 70-77 (*p* < 0.0001) and CT584 154-164 (*p* < 0.0001) as measured by ELISA. End point dilution was defined as the reciprocal of the highest sample dilution with an OD450 reading twice that of blank wells. Horizontal lines represent means +/− SEM.

**Figure 5 vaccines-10-00111-f005:**
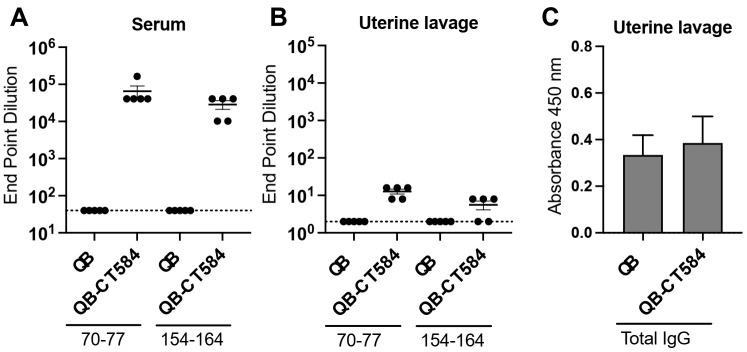
Vaccine epitope-specific antibody is detectable at low levels in uterine lavage. Serum (**A**) and lavage of the lumen of the uterus (**B**,**C**) was collected from mice immunized with either Qβ-CT584 or the negative control Qβ alone. (**A**,**B**) IgG specific to vaccine epitopes CT584 70–77 and CT584 154–164 was measured by ELISA. End point dilution was defined as the reciprocal of the highest sample dilution with an OD450 reading twice that of blank wells. Antigen-specific IgG was detectable in uterine lavage (**B**) but at lower levels than in serum (**A**). (**C**) The amount of total IgG in uterine lavage was comparable between the two vaccination groups.

**Figure 6 vaccines-10-00111-f006:**
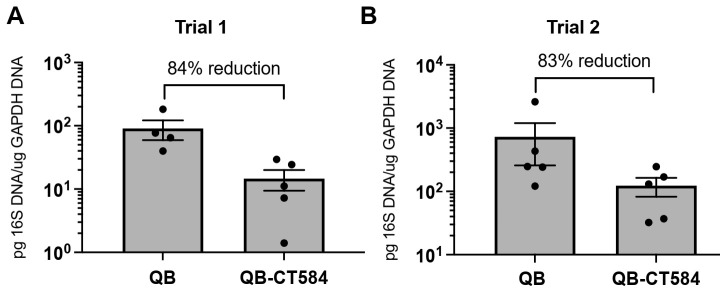
Pre-incubation of *C. trachomatis* with immune serum from Qβ-CT584-vaccinated mice reduces infection in the upper genital tract. *C. trachomatis* were pre-opsonized in vitro with heat-inactivated serum pooled from mice immunized with Qβ-CT584 or Qβ alone, immediately prior to transcervical inoculation into the upper genital tract of naïve female mice. Bacterial burden in the upper genital tract was measured by qPCR. Data are presented from independent pre-opsonization experiments using immune serum from two independent vaccine trials, Trial 1 ((**A**), *p* = 0.0159) and Trial 2 ((**B**), *p* = 0.095). Pre-incubation with Qβ-CT584 serum resulted in a 83–84% reduction in bacterial burden compared to pre-incubation with negative control Qβ serum. Horizontal lines represent means +/− SEM.

## Data Availability

The data presented in this study are available within the paper and on request from the corresponding authors.

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
