# Peer review of "Immunogenicity and Protective Capacity of a Virus-like Particle Vaccine against Chlamydia trachomatis Type 3 Secretion System Tip Protein, CT584"

_vaccines, 2022, doi:10.3390/vaccines10010111_

Round 1

Reviewer 1 Report

This work evaluated the in vivo immunogenicity and efficacy of a Chlamydia trachomatis vaccine made of CT584 epitopes coupled to VLP. Among other results, it is reported that the immunization led to a 3-log rise in IgG in serum and uterine homogenates, and passive protection was accomplished after immune sera were pre-incubated with C. trachomatis before inoculating into the genital tract. Additionally, VLPs seem to be an effective novel platform for C. trachomatis vaccines.

The work is well performed, and it is relevant, since this infection is increasing and it is extremely difficult to control. However, the MS has some point that need to be improved:

Introduction:

  • It would be important to indicate world rated of the infection, not only in USA;

M&M:

  • For every reagent and material, there is the need to be indicated country and manufacturer. Several are lacking this information. Please check and correct the entire MS;
  • Why use C57BL/6 mice? Why not BALB/c or other strain? Justify this in the MS;
  • Food and drink ad libitum or other way? Clarify this in the MS;
  • “All animals were monitored for adverse reactions to the vaccine and none were observed.” – what adverse reaction? “none was observed” is a result, not a m&m;\
  • This part is confusing: “The upper genital tract was excised and mechanically homogenized for quantification of bacterial burden. Uterine lavage was collected from exsanguinated female Balb/c mice housed at the University of New Mexico and immunized as described above. The uterus was excised, washed externally in 1mL of PBS, and 200uL of PBS was instilled into the cervical os and collected via uterine horns” - C57BL/6 and BALB/c? This paragraph needs to be clarified;
  • “...were heat-inactivated for 30 minutes at 56C” – 56º;
  • “...incubated at 37C...” – 37º;

Results:

  • “Urogenital C. trachomatis infection in women does not elicit antibody to Qβ-CT584 vaccine epitopes” – remove the italic for “C. trachomatis” (same for other cases, p.e. section 3.4.Check the entire MS);
  • Figure 2: the graph should be taller (yy axis) to allow a better understanding of the number of animals in each condition;

Discussion:

  • This section is too long and it should be reduced. A final conclusion should be interesting.

Author Response

Reviewer #1

This work evaluated the in vivo immunogenicity and efficacy of a Chlamydia trachomatis vaccine made of CT584 epitopes coupled to VLP. Among other results, it is reported that the immunization led to a 3-log rise in IgG in serum and uterine homogenates, and passive protection was accomplished after immune sera were pre-incubated with C. trachomatis before inoculating into the genital tract. Additionally, VLPs seem to be an effective novel platform for C. trachomatis vaccines.

The work is well performed, and it is relevant, since this infection is increasing and it is extremely difficult to control. However, the MS has some point that need to be improved:

We appreciate the reviewer’s positive feedback and specific criticisms, which are all addressed below.

Introduction:

It would be important to indicate world rated of the infection, not only in USA;

Global C. trachomatis incidence rates have been added.

 M&M:

For every reagent and material, there is the need to be indicated country and manufacturer. Several are lacking this information. Please check and correct the entire MS;

The manufacturer and country is now provided for every reagent and material.

Why use C57BL/6 mice? Why not BALB/c or other strain? Justify this in the MS;

We use C57Bl/6 mice in accordance with previous studies that use this strain when modeling C. trachomatis female upper genital tract infection: e.g. C57Bl/6 mice were used to develop the transcervical C. trachomatis infection model (Gondek et al., JI, 2012; referenced in the manuscript) and we previously published using C57Bl/6 in this model (Lijek et al., PNAS, 2018; referenced in the manuscript). C57Bl/6 have also been shown to be superior to other mouse strains in mounting an antibody response to C. trachomatis peptide antigens (Qu et al., Vaccine, 1994; Motin et al., CDLI, 2020). We have added this justification to the manuscript.

Food and drink ad libitum or other way? Clarify this in the MS;

Food and drink were provided ad libitum, which is now specified in the manuscript.

“All animals were monitored for adverse reactions to the vaccine and none were observed.” – what adverse reaction? “none was observed” is a result, not a m&m;\

This sentence has been removed from the materials and methods. We simply meant that the overall health and well-being of the research animals was monitored throughout the study as is standard practice for animal research and required by the IACUC protocol.

This part is confusing: “The upper genital tract was excised and mechanically homogenized for quantification of bacterial burden. Uterine lavage was collected from exsanguinated female Balb/c mice housed at the University of New Mexico and immunized as described above. The uterus was excised, washed externally in 1mL of PBS, and 200uL of PBS was instilled into the cervical os and collected via uterine horns” - C57BL/6 and BALB/c? This paragraph needs to be clarified;

This section has been revised and separated into two paragraphs to clarify the two different sample collection methods (uterine homogenate vs uterine lavage).

“...were heat-inactivated for 30 minutes at 56C” – 56º; “...incubated at 37C...” – 37º

Degree symbols added.

Results:

“Urogenital C. trachomatis infection in women does not elicit antibody to Qβ-CT584 vaccine epitopes” – remove the italic for “C. trachomatis” (same for other cases, p.e. section 3.4.Check the entire MS);

Italics have been corrected.

Figure 2: the graph should be taller (yy axis) to allow a better understanding of the number of animals in each condition;

The number of human serum samples included in each condition has been added to the text in the section “Human serum samples.”

Discussion:

This section is too long and it should be reduced. A final conclusion should be interesting.

We have shortened the discussion section in response to both reviewers’ requests. Please see our line-by-line response to Reviewer 2’s comments for a detailed list of changes to the discussion section.

Reviewer 2 Report

The authors present an interesting method utilizing a VLP vaccine approach for displaying the tip protein (CT584) of the C. trachomatis T3SS needle complex to the immune system. Female mice were immunized intramuscularly, challenged transcervically with C. trachomatis, assessed for systemic and local antibody responses and bacterial burden in the upper genital tract. Immunization resulted in a 3-log increase in epitope-specific IgG in serum. While IgA levels were not investigated specifically, uterine lavage did demonstrate that measurable levels of mucosal antibodies are present to the target antigens.

The authors note that sera from women infected with C. trachomatis and virgin controls had low titers to CT584 epitopes, and suggest that this may indicate these may be cryptic epitopes during natural infection that can be rendered immunogenic by their VLP platform. They note that C. trachomatis burden in the upper genital tract of mice varied after active immunization, but they were able to demonstrate that sera from immunized mice when pre-incubated with C. trachomatis prior to inoculation into the genital tract reduced measurable bacterial counts 3 days post challenge.

Overall the study was well-conducted, and the results, for the most part, are presented well. However, despite their relatively sound scientific approaches and well-designed study, the authors over-interpret their results in multiple places throughout the manuscript in both the results and discussion sections. This occurs so frequently that it has risen to the level of a major issue that must be addressed prior to acceptance of the manuscript. Additionally, there are an additional few points the authors should address concerning the interpretation of their results as well as the justification for their vaccination approach.

Major Comments:

  • The authors speculate that the reason sera from women infected with trachomatis had low titers to CT584 epitopes was because the epitopes are cryptic during natural infections. They do not address alternative explanations, the two most prominent being 1) antibodies produced to relevant antigens conducive to protection are more plentiful in the mucosa (IgA) than serum (IgG) and 2) intramuscular immunization does not induce a comparable immune response to C. trachomatis infecting a mucosal membrane or an intranasal route of immunization.

Figure 4c:

  • If the authors wish to make this association then they should provide R2 confidence values for their linear regression, so that the reader can appreciate how well the data fit a linear regression (or not). They should also conduct the same linear regression and analysis for their QB control group, as there appear to be a few mice that demonstrated measurable (non-zero) titers. From the data points mapped in 4c, the negative slop appears to be largely the result of the single outlier in the 104 group.

Figure 6:

  • It is unclear why it was necessary to carry out this experimentation in vivo. The authors could have just as easily demonstrated the neutralizing capability of their serum in vitro using cell monolayers. The only conceivable benefit of an in vivo experiment would be if the infection was conducted as a time course study, demonstrating that neutralization by CT584-specific antibodies altered the course of the infection by decreasing bacterial burden over the entirety of an infection and/or time to clearance, rather than just demonstrating an initial decrease in bacterial load due to neutralizing antibodies reducing the initial colonization numbers.
  • That said, the authors do demonstrate that trachomatis incubated with CT584- specific antibodies is less effective at colonizing the upper genital tract, reducing bacterial burden by a little less than 1 log. It is therefore likely that CT584-specific antibodies have the potential to be neutralizing. However, as the authors only looked at a single time point (3 days post infection), it is unclear whether this level of bacterial reduction early on in infection is meaningful or if once the C. trachomatis infection is established the QB-CT584 serum pre-incubated group effectively ‘catches up’ to the negative control.

  • Line 412-414: This summary is misleading and an over-interpretation of the results presented in the article.
  1. is not supported by experiments carried out in Figure 4a. If the authors wish to make this claim, they need to reconcile their results presented in Figure 3b and Figure 4a. Either pool all of the data from the two studies and conduct an appropriate statistical analysis or conduct the previously mentioned controls in the analysis of Figure 4c
  2. While it can be argued that antibody neutralization works differently in vivo than it does in vitro, it is the opinion of this reviewer that the use of mice for conducting what amounts to a Chlamydia invasion neutralization assay utilizing non-physiological levels of antigen-specific serum should not be celebrated or held out as an example for future studies.
  3. nothing in the manuscript demonstrates an ability of the CT584 to ‘block infection’; a more appropriate assay for demonstrating that CT584-specific neutralizing antibodies ‘block’ cell entry by C. trachomatis EBs would have been an in vitro cell invasion assay carried out utilizing standard inclusion-forming unit (ifu) counts

  • The field of Chlamydia vaccinology has long fought over which immunological response (humoral or cell-mediated) is more important for conferring long-term protection. The current, shaky middle ground is ... both. In their manuscript, the authors focus exclusively on the humoral response to their vaccine and completely avoid any mention at all of the potential for cell-mediated response. This needs to be addressed. The authors need to state why their approach is valid and likely to be beneficial in the future and worth further development. If they do not address the cell mediated response to their vaccine in their results section, then they need to state why.

Minor comments:

  • Line 61-66: It is important to point out that neutralizing antibodies were only accessed using serum IgG. MOMP-specific IgG and IgA were considerably less prevalent in mucosal secretions, and were not shown to be neutralizing.

  • Figure 4: It is unfortunate that the statistical significance did not hold up in the second challenge study. Did the author’s see any differences in the number of bacteria in their starting inoculums between the two challenges? The author’s may consider mentioning that the failure to achieve colonization in 4 of their QB control animals likely affected their results. Additionally, given the level of CT584-specific antibody generated in immunized mice, the vaccine may have demonstrated effectiveness in time-to-clearance. 3 days may be insufficient for this potential phenotype to be investigated, however, the authors would need to better describe the course of an infection in their transcervical-infection model in order to better interpret this result.

  • Intramuscular immunizations are likely not ideal for inducing an immune response targeting a pathogen that spends the entirety of the extracellular portion of its developmental cycle in the mucosa of the genital tract.

  • Lines 383-385: Without the requested additions to this analysis, this statement is not supported by the author’s results. Additionally, as bacterial burden was measured at only a single time point relatively close to the initial infection day (3 days post infection), the authors should be wary of over interpreting this result.

  • Line 417-420: Given that the authors found i) no significant difference in C. trachomatis burden between their vaccine and control groups (Figure 4a) and ii) the levels of uterine antibody present in the lavages were barely above the assay’s limit of detection, I would not point to their study as proof that intranasal immunization is not required for mounting a significant and protective immunological response to C. trachomatis infections.

Author Response

Reviewer #2

The authors present an interesting method utilizing a VLP vaccine approach for displaying the tip protein (CT584) of the C. trachomatis T3SS needle complex to the immune system. Female mice were immunized intramuscularly, challenged transcervically with C. trachomatis, assessed for systemic and local antibody responses and bacterial burden in the upper genital tract. Immunization resulted in a 3-log increase in epitope-specific IgG in serum. While IgA levels were not investigated specifically, uterine lavage did demonstrate that measurable levels of mucosal antibodies are present to the target antigens.

The authors note that sera from women infected with C. trachomatis and virgin controls had low titers to CT584 epitopes, and suggest that this may indicate these may be cryptic epitopes during natural infection that can be rendered immunogenic by their VLP platform. They note that C. trachomatis burden in the upper genital tract of mice varied after active immunization, but they were able to demonstrate that sera from immunized mice when pre-incubated with C. trachomatis prior to inoculation into the genital tract reduced measurable bacterial counts 3 days post challenge.

Overall the study was well-conducted, and the results, for the most part, are presented well. However, despite their relatively sound scientific approaches and well-designed study, the authors over-interpret their results in multiple places throughout the manuscript in both the results and discussion sections. This occurs so frequently that it has risen to the level of a major issue that must be addressed prior to acceptance of the manuscript. Additionally, there are an additional few points the authors should address concerning the interpretation of their results as well as the justification for their vaccination approach.

We appreciate the reviewer’s very thoughtful critique of the manuscript, their positive feedback on the study design, and their constructive criticism about over-interpretation. We do not wish to over-state our results and have made significant changes as requested to temper the language in the manuscript to better align the text with the dataset, see line by line responses below.

Major Comments:

The authors speculate that the reason sera from women infected with trachomatis had low titers to CT584 epitopes was because the epitopes are cryptic during natural infections. They do not address alternative explanations, the two most prominent being 1) antibodies produced to relevant antigens conducive to protection are more plentiful in the mucosa (IgA) than serum (IgG) and 2) intramuscular immunization does not induce a comparable immune response to C. trachomatis infecting a mucosal membrane or an intranasal route of immunization.

The reviewer rightly points out that there are other interpretations of these data, which we have now added to the discussion section (see text below for convenience). We also removed mention of cryptic epitopes from the abstract to avoid over-speculation.

New discussion text: “Whether CT584 epitopes 70-77 and 154-164 are cryptic epitopes remains to be determined: natural infection in women does not produce robust serum IgG but these systemic titers may not be representative of mucosal antibody titers. Immunity to mucosal infection and/or immunization is likely to differ from intramuscular immunization with these peptides, and there may also be species-specific differences in immunogenicity between humans and mice. Future studies of the genital mucosal immune response to CT584 epitopes in humans and mouse models are warranted.”

Figure 4c:

If the authors wish to make this association then they should provide R2 confidence values for their linear regression, so that the reader can appreciate how well the data fit a linear regression (or not). They should also conduct the same linear regression and analysis for their QB control group, as there appear to be a few mice that demonstrated measurable (non-zero) titers. From the data points mapped in 4c, the negative slop appears to be largely the result of the single outlier in the 104 group.

We have removed this figure panel and all associated text to avoid over-interpretation. The R2 confidence values are unconvincing and the trend is not statistically significant.

Figure 6:

It is unclear why it was necessary to carry out this experimentation in vivo. The authors could have just as easily demonstrated the neutralizing capability of their serum in vitro using cell monolayers. The only conceivable benefit of an in vivo experiment would be if the infection was conducted as a time course study, demonstrating that neutralization by CT584-specific antibodies altered the course of the infection by decreasing bacterial burden over the entirety of an infection and/or time to clearance, rather than just demonstrating an initial decrease in bacterial load due to neutralizing antibodies reducing the initial colonization numbers.

We chose to perform an in vivo infection rather than an in vitro neutralization assay in an effort to maximize physiological relevance of our findings and to better allow the reader to compare these results with the other in vivo infections in the manuscript. The mucosal barrier of the genital tract includes more than a homogenous, immortalized cell monolayer; C. trachomatis must traverse mucus and its associated defense molecules and avoid phagocytes, which we previously showed are present in the uterus at this time post transcervical infection with this serovar (Lijek et al., PNAS, 2018). For reference and to address questions about the model below, that paper shows a time course of C. trachomatis serovar D burden in this upper genital tract model (Figure 1, panel D).

Choosing an in vivo infection vs an in vitro neutralization assay also allows us to remain agnostic about how antibody-mediated protection against C. trachomatis might function. Protective antibody could function by neutralizing the bacteria (hindering attachment or critical virulence factors, etc.) or through enhancing opsonophagocytic clearance of the bacteria. Using an in vivo protection assay allows us to be inclusive of both possibilities, since an in vitro neutralization assay on cell monolayers would only assess neutralization and not opsonophagocytosis. We hope to tease apart more of this mechanism in future studies beyond the scope of this manuscript. A brief discussion of this point has been added to the discussion.

We considered a time course study, as the reviewer mentions, but ultimately decided against it for a few reasons. We reasoned that pre-incubation with a (hypothetically protective) antibody would only be capable of altering the outcome of the initial round of infection, since the antibody source is only present on the initial inoculum (from the ex vivo incubation with serum). Once C. trachomatis had successfully infected the uterine epithelium and replicated, the resulting second generation of Chlamydia EBs would not have a new source of antibody in vivo. So we predicted it was unlikely to see any passive antibody-mediated differences in bacterial burden at later times post infection. Also, quantifying C. trachomatis burden in the upper genital tract requires sacrificing the mice, so a time course experiment would require a much larger number of mice and would not be a true, continuous time course since the individual mice would change between each time point. Given the variability in the C. trachomatis burden already inherent in the model, it seemed unlikely to be a productive or ethical use of mice. Therefore, we were pleased to see differences in bacterial burden at the day 3 time point we chose. In future experiments beyond the scope of this manuscript, we might investigate the impact of antibody on time-to-clearance using the C. muridarum model which allows for vaginal shedding to be measured continuously over time, which we now mention in the discussion section.

That said, the authors do demonstrate that trachomatis incubated with CT584- specific antibodies is less effective at colonizing the upper genital tract, reducing bacterial burden by a little less than 1 log. It is therefore likely that CT584-specific antibodies have the potential to be neutralizing. However, as the authors only looked at a single time point (3 days post infection), it is unclear whether this level of bacterial reduction early on in infection is meaningful or if once the C. trachomatis infection is established the QB-CT584 serum pre-incubated group effectively ‘catches up’ to the negative control.

Previous use of this model in our hands and by others shows that the bacterial burden peaks around day 3 and then declines over time (e.g. again, see Lijek et al., PNAS, 2018 Figure 1D, and Gondeck et al., JI, 2012, both referenced in the manuscript). This is why we chose the day 3 time point to quantify C. trachomatis in vivo in our experiments: it is at the expected peak of bacterial burden in this model; it comes after approximately one round of bacterial replication; and is late enough that any of the non-productive inoculum would likely be cleared and so represents “true” infection levels (rather than infection + residual inoculum which might be expected at day 1).  We also have no reason to believe that the rate of replication of bacteria incubated with immune serum would surpass (or differ in any way) from the rate of replication of the bacteria replicated with control serum, so we do not expect a “catching up” to occur. However, these are speculations that we did not test in the present study, and so we have added to the manuscript a discussion of future directions such as performing a time-to-clearance experiment.

Line 412-414: This summary is misleading and an over-interpretation of the results presented in the article.

  1. is not supported by experiments carried out in Figure 4a. If the authors wish to make this claim, they need to reconcile their results presented in Figure 3b and Figure 4a. Either pool all of the data from the two studies and conduct an appropriate statistical analysis or conduct the previously mentioned controls in the analysis of Figure 4c
  2. While it can be argued that antibody neutralization works differently in vivo than it does in vitro, it is the opinion of this reviewer that the use of mice for conducting what amounts to a Chlamydia invasion neutralization assay utilizing non-physiological levels of antigen-specific serum should not be celebrated or held out as an example for future studies.
  3. nothing in the manuscript demonstrates an ability of the CT584 to ‘block infection’; a more appropriate assay for demonstrating that CT584-specific neutralizing antibodies ‘block’ cell entry by C. trachomatis EBs would have been an in vitro cell invasion assay carried out utilizing standard inclusion-forming unit (ifu) counts

These lines have been removed. The phrase “block infection” has been removed throughout the manuscript (and replaced with “inhibit infection” or “reduce bacterial burden” etc.), as we did not mean to imply a specific neutralization mechanism with the word “block”, but rather a general comment that infection had been reduced.

The field of Chlamydia vaccinology has long fought over which immunological response (humoral or cell-mediated) is more important for conferring long-term protection. The current, shaky middle ground is ... both. In their manuscript, the authors focus exclusively on the humoral response to their vaccine and completely avoid any mention at all of the potential for cell-mediated response. This needs to be addressed. The authors need to state why their approach is valid and likely to be beneficial in the future and worth further development. If they do not address the cell mediated response to their vaccine in their results section, then they need to state why.

We focus on antibody-mediated immunity in this manuscript because 1) previous studies of QB VLP vaccines show them to be a robust platform for stimulating antibody and 2) we designed the vaccine to include B-cell epitopes but made no attempt to include T-cell epitopes. Our vaccine couples the QB-VLP to short peptides identified by IEDB to be ideal B-cell epitopes for our antigen, CT584. These short peptides are likely too small to be properly processed for display on MHC. Because this vaccine was not designed to induce a robust or protective cell mediated response against CT584, this aspect was not explored in this study.

We completely agree with the reviewer that it is likely that both humoral and cell-mediated immunity has a valuable role to play in protecting against Chlamydia, and that an ideal Chlamydia vaccine would stimulate both arms of the immune response. As we mention below, we suspect that an effective immunization scheme may benefit from a combination of administration routes and/or antibody- and T-cell stimulating features. Peptide-conjugated VLPs, like we explore in this paper and if coupled to the “right” choice of peptides, may be a useful antibody-stimulating component for a future heterologous approach to Chlamydia vaccines. 

As requested, we have added this discussion to the manuscript.

Minor comments:

Line 61-66: It is important to point out that neutralizing antibodies were only accessed using serum IgG. MOMP-specific IgG and IgA were considerably less prevalent in mucosal secretions, and were not shown to be neutralizing.

We have added the reviewer’s comments to the text, which now reads (bold for emphasis on new text):

“In a clinical trial, the MOMP-VD4 CTH522 vaccine was shown to be safe and immunogenic in 15 women, inducing neutralizing antibody in serum after three intramuscular injections, though MOMP-specific IgG and IgA were less prevalent in mucosal secretions and not shown to be neutralizing [8].”

Figure 4: It is unfortunate that the statistical significance did not hold up in the second challenge study. Did the author’s see any differences in the number of bacteria in their starting inoculums between the two challenges? The author’s may consider mentioning that the failure to achieve colonization in 4 of their QB control animals likely affected their results. Additionally, given the level of CT584-specific antibody generated in immunized mice, the vaccine may have demonstrated effectiveness in time-to-clearance. 3 days may be insufficient for this potential phenotype to be investigated, however, the authors would need to better describe the course of an infection in their transcervical-infection model in order to better interpret this result.

We were also disappointed that the statistical significance did not hold up in the second challenge study. There was no difference in the starting inoculums nor any other methodological inconsistency that we could identify that might explain the increased variability. We have added the following sentence to the Figure 4 results section as suggested: “The failure to achieve infection in 4 of the QB-immunized control animals may have contributed to these findings.”

The reviewer suggests a logical future experiment that we too have considered: active immunization followed by a time course of bacterial burden. As mentioned previously, the C. trachomatis upper genital tract model requires sacrifice to enumerate bacteria and so burden cannot be measured continuously over time within the same animal. So to perform the time-to-clearance experiment, we would switch to a C. muridarum lower genital infection model where bacterial shedding can be continuously monitored over time within the same animal. We may pursue this line of experimentation in the future and believe it to be beyond the scope of this manuscript on C. trachomatis upper genital tract infection.

Intramuscular immunizations are likely not ideal for inducing an immune response targeting a pathogen that spends the entirety of the extracellular portion of its developmental cycle in the mucosa of the genital tract.

This may well prove true for Chlamydia vaccines. Intramuscular immunization with Gardasil and Cervarix (which are VLP vaccines that protect via antibody) has been highly effective at preventing mucosal genital tract infection with HPV and its disease sequelae. And there is other precedent for effective parenteral vaccines against mucosal respiratory tract pathogens like S. pneumoniae and influenza. Immunity in the genital tract could potentially be enhanced by using a mucosal immunization strategy, or perhaps even a heterologous “prime-pull” approach as is being pursued for HSV (e.g. Berstein et al., npj Vaccines, 2019). We’re excited by the idea of these novel approaches to Chlamydia vaccination and suspect that an effective immunization scheme may benefit from a combination of administration routes and/or antibody- and T-cell stimulating features.

Lines 383-385: Without the requested additions to this analysis, this statement is not supported by the author’s results. Additionally, as bacterial burden was measured at only a single time point relatively close to the initial infection day (3 days post infection), the authors should be wary of over interpreting this result.

We intended these lines to summarize the work of others, not our own results. We were also remiss in over-simplifying the results of Gondek et al., JI, 2012 (ref #35). So to clarify and add more context, we have changed the text as shown below (new text is bolded, removed text is strikenthrough).

“Similarly, antibody elicited by natural infection has been shown to be dispensable during the clearance of a primary upper genital tract infection with C. trachomatis in mice [35] and in women does not correlate with re-infection outcomes [12]. Yet, others have shown that protective antibody responses superior to those from natural infection can be achieved exogenously through monoclonal antibody treatments or experimental vaccines, as with the HPV vaccines [10,40–42]. Our data add to this line of evidence by demonstrating that pre-incubation with high-titer Qβ-CT584 immune serum is sufficient to reduce C. trachomatis burden in the upper genital tract. To our knowledge, Qβ-CT584 is the only vaccine besides CTH522, which is currently in clinical trials, to elicit antibody capable of passively inhibiting C. trachomatis infection in the upper genital tract of mice [10].

Line 417-420: Given that the authors found i) no significant difference in C. trachomatis burden between their vaccine and control groups (Figure 4a) and ii) the levels of uterine antibody present in the lavages were barely above the assay’s limit of detection, I would not point to their study as proof that intranasal immunization is not required for mounting a significant and protective immunological response to C. trachomatis infections.

This sentence has been removed:

“They also demonstrate that the production of serum and uterine antibody does not require intranasal immunization as in previous studies and can be achieved after intramuscular immunization with the appropriate platform, such as a VLP.”

Round 2

Reviewer 2 Report

The authors have addressed all of my concerns. Again, a very well-designed study. I look forward to future work utilizing the approaches they describe.